# Affordable RFID loggers for monitoring animal movement, activity, and behaviour

Natasha Dean Harrison[1]*, Ella L. Kelly[2]

**1** School of Biological Sciences, University of Western Australia, Crawley, Western Australia, Australia,
**2** School of BioSciences, University of Melbourne, Parkville, Victoria, Australia

* natasha.harrison@research.uwa.edu.au

## Abstract

Effective conservation management strategies require accurate information on the movement patterns and behaviour of wild animals. To collect these data, researchers are increasingly turning to remote sensing technology such as radio-frequency identification (RFID). RFID technology is a powerful tool that has been widely implemented in ecological research to identify and monitor unique individuals, but it bears a substantial price tag, restricting this technology to generously-funded disciplines and projects. To overcome this price hurdle, we provide detailed step-by-step instructions to source the components for, and construct portable RFID loggers in house, at a fraction of the cost (~5%) of commercial RFID units. Here, we assess the performance of these RFID loggers in the field and describe their application in two studies of Australian mammal species; monitoring nest-box use in the Northern quolls (*Dasyurus hallucatus*) and observing the foraging habits of quenda (*Isoodon fusciventer*) at feeding stations. The RFID loggers performed well, identifying quenda in >80% of visits, and facilitating the collection of individual-level behavioural data including common metrics such as emergence time, latency to approach, and foraging effort. While the technology itself is not novel, by lowering the cost per unit, our loggers enabled greater sample sizes, increasing statistical power from 0.09 to 0.75 in the quoll study. Further, we outline and provide solutions to the limitations of this design. Our RFID loggers proved an innovative method for collecting accurate behavioural and movement data. With their ability to successfully identify individuals, the RFID loggers described here can act as an alternative or complementary tool to camera traps. These RFID loggers can also be applied in a wide variety of projects which range from monitoring animal welfare or demographic traits to studies of anti-predator responses and animal personality, making them a valuable addition to the modern ecologists' toolkit.

**Data Availability Statement:** Data from the quoll and quenda studies can be found on the following GitHub repositories: github.com/elkelly/RFIDlogger and https://github.com/natasha-harrison/Woylie respectively.

## Introduction

Effective conservation management strategies require accurate information on the movement patterns and behaviour of wild animals [1–4]. Collection of these data is made difficult by the wide-ranging movements of many species, the hostile habitats in which they can live, and the

**Funding:** This work was supported by the Australian Research Council (LP150100722; FT160100198 to A/Prof Ben Phillips and A/Prof Jonathan Webb); Australian Commonwealth Government RTP Scholarship (to N.D.H); Margaret Middleton Fund Award for Endangered Australian Native Vertebrate Animals (to E.K); Hermon Slade Foundation (HSF21054 to Nicola Mitchell); and Holsworth Wildlife Research Endowment (to E.K and N.D.H). The funders had no role in study design, data collection and analysis, decision to publish, or preparation of the manuscript.

**Competing interests:** The authors have declared that no competing interests exist.

potential for human presence to alter or confound the natural behaviour of individuals [5, 6]. To overcome these challenges, researchers are increasingly turning to remote sensing technology that removes the need for a human observer, for example, camera traps, accelerometers, and radio tagging [7, 8]. One such technology, is radio-frequency identification (RFID), which has long been utilised to uniquely identify individuals across many disciplines [9].

RFID technology uses electromagnetic fields to detect unique radio tags, which can be inserted into animals in the form of tiny passive integrated transponder (PIT) tags, also known as microchips [10]. PIT tags are a good alternative to other marking techniques, such as ear tags, as they are retained better, and can be used to permanently tag animals from all taxa including mammals [11], birds [12], fish [13], reptiles [10] and amphibians [14]. As a result, numerous studies of wildlife have adapted RFID technology to individually identify animals and monitor their movements [15]. For example, through the implementation of RFID technology, Boarman and colleagues determined the frequency of highway underpass use by desert tortoises [16], Skov and colleagues monitored seasonal dispersal in fish [17], and Bandivadekar and colleagues evaluated feeder visits in hummingbirds [18]. Despite its extensive use, the major limitation of RFID technology is that it is expensive [10, 19], making it inaccessible to areas of study with limited funds, such as conservation [20], and restricting projects to small sample sizes [21].

In this study, we provide detailed instructions to build short-range RFID loggers from individual components, substantially reducing the costs per unit compared to commercial alternatives by up to 96%. A commercial RFID logger capable of reading and storing individual identities costs between $1800 and $3000 (AUD) (e.g. Microchips Australia: LID650 reader & ANTSQR300 antenna quoted between $1800–2000 AUD), and we purchased the components for our loggers for $130 (AUD), a mere 4–7% of this price. These data-logging stations record each individuals' unique identity and a timestamp, making them capable of measuring popular behavioural metrics (such as emergence time, latency to approach, and foraging effort) and completely removing the need for a human observer. Such metrics, and others made possible by our RFID loggers, can give insights into demography, perceived predation risk and animal personality [22, 23], however, the feasibility of collecting such metrics and the performance of the RFID loggers themselves requires field-testing.

We validated the design for our RFID loggers in two studies of Australian mammals; monitoring nest-box use by Northern quolls (*Dasyurus hallucatus*) in captivity and observing the foraging habits of quenda (South western brown bandicoots; *Isoodon fusciventer*) at feeding stations in an urban reserve. Using self-built loggers reduced the cost of RFID equipment from $40,000 to $2,600 and $10,000 to $650 in the quoll and quenda studies respectively. Here we evaluated the performance (% of PIT-tagged individuals successfully identified by the RFID loggers) and tested the battery life of our RFID loggers. Furthermore, we established the feasibility of collection of four common behavioural measurements; nest-box emergence time, nest-box activity, latency to approach feeding station, and foraging effort at feeding station. Finally, we discuss the limitations of this methodology, provide suggestions for future developments, and identify additional uses for this technology within ecological research.

## Materials, methods and application

### Building RFID loggers

RFID loggers require three main components; the PIT tags/microchips for each animal, the RFID logger, and the antenna. We used the RFIDlog by Priority 1 Designs (Melbourne, Australia), with a 16cm antenna. This particular logger exclusively records PIT tags that operate at a frequency of 134.2khz (conforming to the ISO 11784 and ISO 11785 standards in Australia;

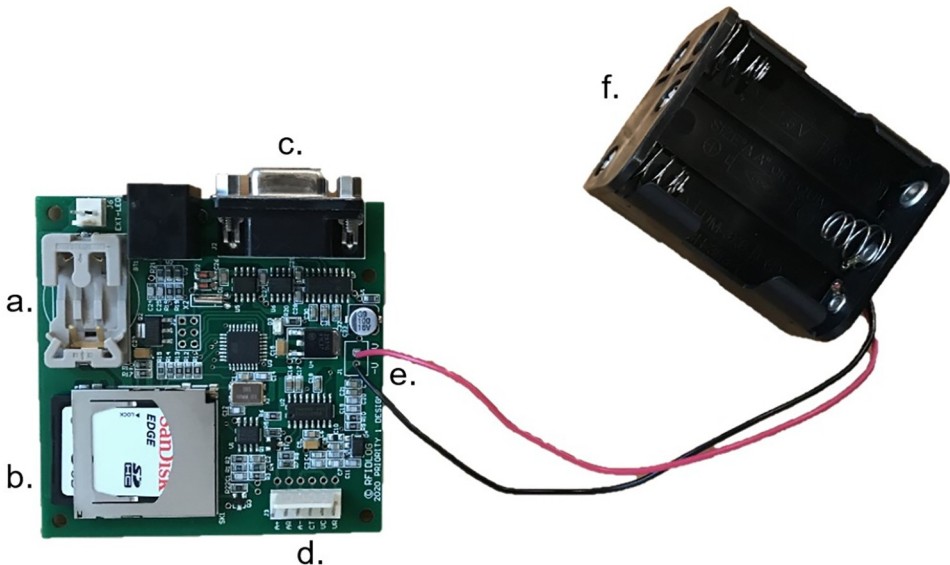

**Fig 1. RFID logger (Priority 1 Designs; Melbourne, Australia).** Components include battery casing for a CR2032 3V lithium that powers the units internal clock (a), SD card for storing RFID reads (b), serial port DBC computer connection (c), external antenna port (d), and an external battery pack for a power source to support unit (f), soldered directly to the power input (e).

[24]). For use in small mammals, these PIT tags are inserted between the shoulder blades, on the back of the animal [25]. The RFID logger requires an external power source, for which we used battery packs, capable of housing 6x AA batteries, which are soldered directly to the printed circuit board of the RFID logger (Fig 1(E) and 1(F)). When scanning, the logger draws 80mAh per hour and one can therefore calculate how much external battery power is needed, for example, using 6x rechargeable NiMH AA batteries (2550mAh) should power the logger for 31 hours continuously (2550/80 = 31.8). The antenna is connected to the RFID logger (at Fig 1(D)) using header crimp pins, allowing it to be easily removed. For a detailed guide to constructing the RFID loggers from components, please see S1 Table. Aside from the RFID logger and antenna components, our field housing designs (see Fig 2) benefit from the use of simple and readily available (local hardware or hobby store) tools and parts. A full list of these can be found in the supplementary materials (S1 File), along with price estimates relevant to the time of publication.

## Application in field studies

**Northern quoll nestbox activity.** The first application of our RFID loggers was part of various research projects investigating methods to mitigate the impact of toxic cane toads on Northern quolls [25, 26], particularly exploring the personalities of different individuals to investigate links between toad and predator response and boldness. Northern quolls (a carnivorous marsupial) were collected from Astell island, Northern Territory (-11.885743, 136.424008) in February 2018 and brought to the Territory Wildlife Park. Here, 20 RFID loggers were deployed on these new arrival's nest boxes (Fig 2) over 3 nights to record latency to emerge from the nestbox (time until first read), as well as the number of movements the quoll made in and out of the nestbox over the first night in captivity (count of reads) from 45 individuals. The University of Melbourne Animal Ethics Committee gave their permission to carry out this experiment (ID number 1413369.2).

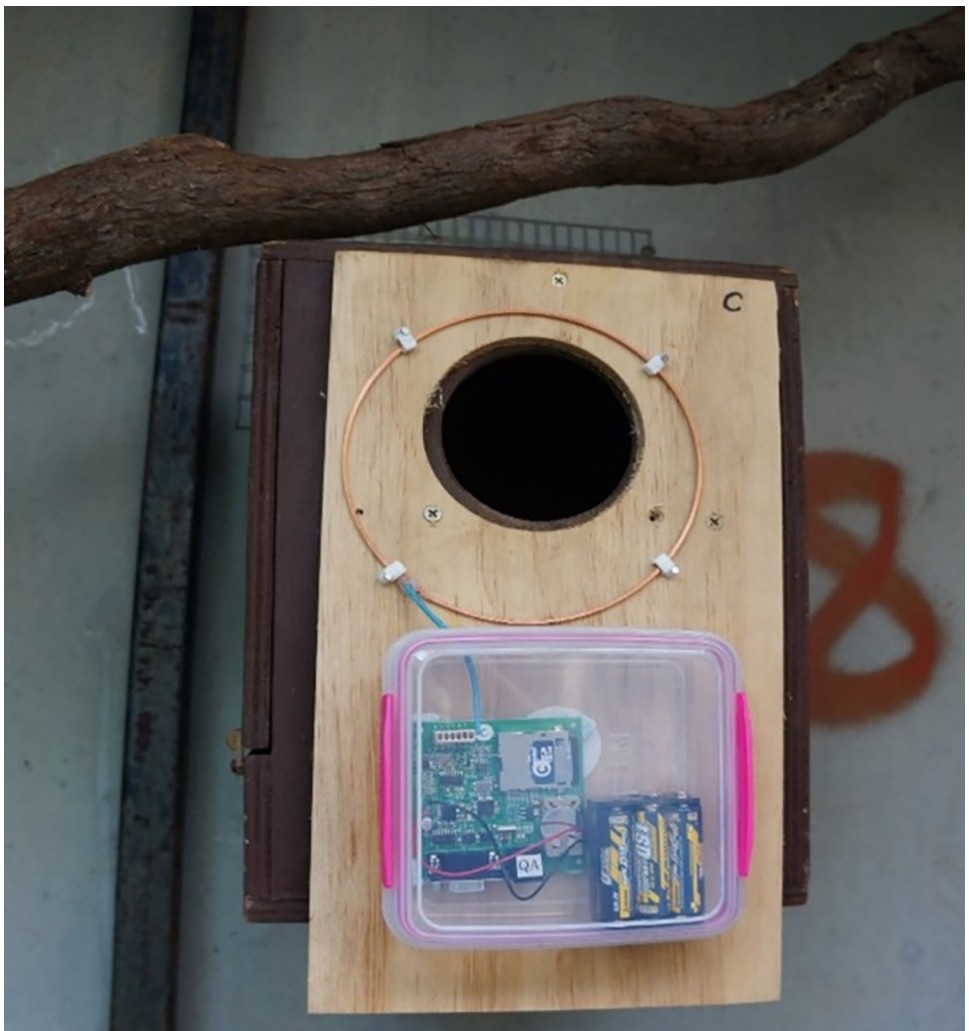

**Fig 2. RFID logger (inside weather-proof housing) attached to a nestbox for Northern quolls with the RFID antenna placed around the entrance.**

**Quenda study.** The second application of our RFID loggers was on a wild population of quenda, a small omnivorous marsupial, residing in Craigie Bushland in Perth, Western Australia (-31.792772, 115.778711). Here, the RFID loggers were used to record the time of approach, foraging effort and individual identity of bandicoots at each feeding station. For this study, the antenna of the RFID logger was fixed around the entrance to a feeding station (a 90˚ PVC storm pipe elbow or straight length of PVC, with a food reward buried at the bottom) (Fig 3), which encourages the animal to pass through the antenna, giving the best probability of detection. Each RFID logger was monitored by a camera trap to validate the data captured by the RFID loggers. Camera traps alone were not sufficient for this study as quenda cannot be individually identified from images. We deployed 5 RFID loggers over 4 nights. From these data, we determined the latency to approach (time of first entry), foraging effort (length of time between first and last entry read) and percentage of quenda successfully identified (validated with camera-trap images). Within individual repeatability of latency to approach and foraging effort was calculated using a linear mixed model repeatability estimate with a restricted maximum likelihood function in the program R [27] using the package rptR [28].

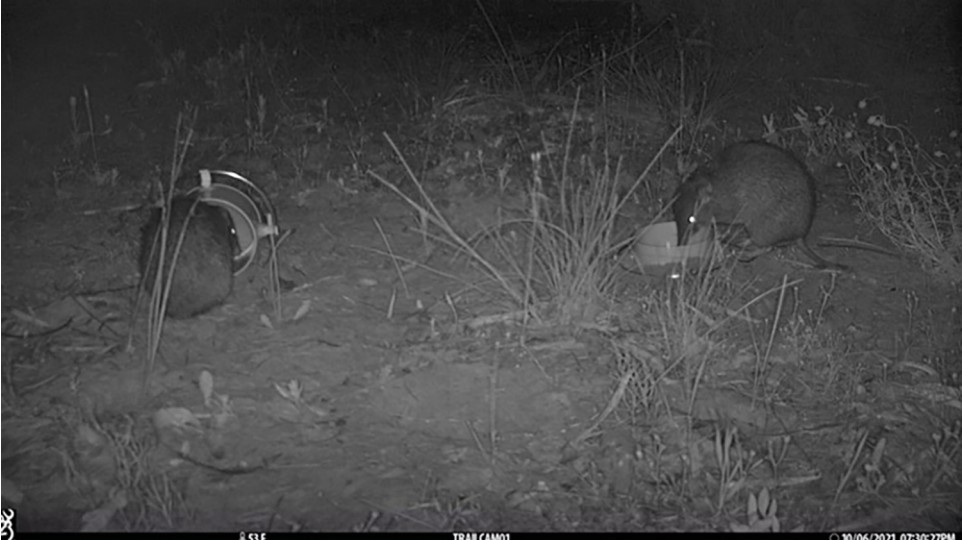

**Fig 3. Quenda entering feeding stations (PVC pipes) through an RFID logger antenna allowing them to be uniquely identified.**

The University of Western Australia's Animal Ethics Committee gave their permission to carry out this study (2021_ET000428_v2).

## System performance and results

### Northern quoll nocturnal activity

The RFID loggers were successful in logging time to emergence and overnight activity of the northern quolls during their first night in captivity. Of the 45 quolls measured, 89% emerged from their nestbox on the first night in captivity. Of these, the average time to emergence (± SEM) was 182.2 ± 35.8 minutes. The quolls that emerged made on average 38.3 ± 13.7 movements in or out of the nestbox, however, we were unable to distinguish between movements to enter or exit the nestbox. There was one record which had 558 reads–a strong outlier compared to the rest of the records. Looking at the raw data, we found that for a period during the experiment the logger was recording multiple records in a minute–potentially as the quoll sat at the entrance to the nestbox. In an attempt to remove this bias, we filtered the data to remove multiple records from the same minute, and computed number of movements again. This corrected the outlier and resulted in much more conservative measures generally, with an average of 4.9 ± 0.84 movements. There was no significance difference in emergence time between the sexes (ANOVA: F = 0.81, p = 0.37; S1 Fig in S4 File). There was also no significant difference between the sexes in the amount of activity recorded entering or exiting the nestbox (ANOVA: F = 0.06, p = 0.82; S2 Fig in S4 File). All RFID loggers functioned for 24 hours as expected.

### Quenda foraging habits

The RFID loggers were successful in identifying individuals, logging the latency to approach feeding stations, and recording data from which foraging effort could be calculated. From 148 total reads, camera traps revealed that the RFID loggers received 31 visits from quendas, and individuals were correctly identified during 25 of these (80% of visits to feeding stations; not all visits resulted in an animal entering the station). The average latency to approach (in minutes after sunset ± SEM) was 121.63 ± 15.51 across individuals. Foraging effort (difference in

time between first and last entry read) ranged from 0.03 minutes to 5.92 minutes, with an average of 1.72 ± 0.35. All animals detected were female so we were unable to test for sex differences. Neither latency to approach feeding stations, nor foraging effort at feeding stations were repeatable within individuals (R = 0 ± 0.09 and R = 0 ± 0.08 respectively), though repeat measures were only available from four individuals. All RFID loggers functioned for 24 hours as expected.

## Discussion

Understanding animal behaviours at the individual level is crucial for the implementation of effective conservation management strategies [3, 8], however, the collection of these data can be hindered by cost. We overcome this price hurdle by providing an affordable (5% of traditional commercial models) design assembled from easily accessible components which successfully identified PIT-tagged quolls and quendas and provided data from which useful behavioural metrics could be calculated, including nest-box emergence time, nest-box activity, latency to approach feeding station, and foraging effort at feeding station.

The RFID loggers described here promote reliable and accurate research in ecology. The reduction in cost per unit encourages more robust experimental designs by enabling larger sample sizes and increased replicates through space and time. For example, in the quoll study where nest boxes were simultaneously monitored over three evenings, the price of two commercial units ($3,600 AUD) could alternatively be used to purchase the components for 27 self-built loggers, increasing the sample size from 6 (2 readers over 3 nights) to 81 (27 readers over 3 nights), and improving the power of detecting a subtle difference in emergence time between two groups from 0.09 to 0.75 [29] (S2 File). Further, by recording individual identity, the loggers facilitate the investigation of the repeatability of behaviours within individuals and populations across contexts–fundamental to gaining an understanding of animal personalities but also crucial when validating behavioural measures. Our finding that neither latency to approach, nor foraging effort were repeatable within individual quenda suggests that allocation of time to foraging in quenda is context dependent (e.g. hunger), rather than attributable to personality traits. Our sample size for this analysis, however, was small (n = 4) so we recommend this be tested with more replicates.

Despite their successful implementation, our RFID loggers are not without their limitations. The system presented here is unidirectional, meaning the logger cannot determine the direction of the movement (i.e. entering or exiting of the nestbox). This system can, however, be adjusted to allow directional readings with the addition of an auxiliary RFID logger and antenna (S3 File). The two antennas can then be set up on either side of a tunnel to record (via time differences between antennae) which direction the animal is travelling. In terms of battery life, the RFID loggers lasted 24 hours as expected across both studies. While this was more than enough for these applications (where sites were checked daily), other studies may wish to deploy these loggers for longer and will require longer battery life. In these instances, we suggest using alternate power sources such as SLA batteries or small solar panels. Our results have also shown the importance of testing the accuracy of the RFID readers, which can sometimes fail to read a visit. Although camera trap data was not able to be collected for the quoll study to ground-truth the results, camera trap footage from the Quenda study showed an 80% accuracy rate.

Using the feeding station setup described in the quenda study, the RFID loggers could capture additional behavioural metrics at the individual level such as giving-up-densities (GUDs: a density threshold of foods at which animals cease foraging; [30]), proportion of time allocated to various behaviours (e.g. foraging and vigilance; [31]), or choice experiments (e.g.

predator and control cues at each feeding station [32]). Such metrics can provide valuable insights into animal personalities, perceived risk and anti-predator responses [33, 34], aspects of conservation behaviour that are crucial to implementing effective management strategies [1]. The aforementioned behavioural measures can also be used to evaluate animal welfare, particularly in a captive setting [35, 36].

RFID technology can be instrumental in monitoring populations for demographic studies, giving crucial information on individual survival and dispersal [37]. In the quoll study, we found no sex differences in latency to emerge, nor in quoll activity whilst in captivity. In the wild, however, we may expect to see such sex differences in movement and activity, as males traverse across large home ranges (84 ± 16 ha) in pursuit of females [38]. While camera traps are useful for capture-mark recapture studies where individuals can be uniquely identified visually (e.g. numbats [39]), not all species possess distinct markings that allow them to be recognized. Additional problems can also arise from poor image quality and the misidentification of individuals [40, 41]. RFID loggers are a promising alternative or complement (as in the quenda study) to camera traps. Our RFID loggers performed equally, or better than camera traps in identifying individuals: we were able to successfully identify individuals at 80% of visits compared to camera trap picture detection rates which vary depending on species (e.g. 5.3% of photos of an indistinct deer [42]; 59–80% probability of matching photos of cheetahs [43]; identifications from 73% of detection events of perentie [44]). Further, the RFID loggers described here are cheaper per unit ($130 AUD), compared to camera traps which cost between $300-$1050 AUD (Outdoor Cameras, Australia: Swift Enduro and Reconyx XR6 Ultrafire models respectively).

Our RFID loggers remove the need for a human observer in the field, however, they still require some human effort to collect data (downloaded manually using the Priority 1 Software). Given the rapid development of sensor technology and the modular nature of these RFID loggers, it may be possible to connect these sensors to a mobile network, where data can be received straight into the cloud (the concept of Internet Of Things; IOT [9]). Though this was outside the scope of this particular study, we recommend futures studies considering our RFID loggers take the time to investigate the potential of IOT to increase efficiency in data collection through automation. Such an advancement also opens up many avenues for further application, such as remote trapping of targeted individuals, or access-limited nest boxes based on individual identity.

## Conclusion

The technology we present here is not novel, however, by reducing the cost per unit, our design makes this technology more accessible and facilitates more robust sample designs (larger sample sizes, increased replicates and improved statistical power). The successful implementation of our RFID loggers in the field allowed us to capture common behavioural metrics, and the loggers have the potential to be utilised in pursuit of a broad range of behavioural and demographic questions, making them a valuable tool for use in ecological studies.

## Supporting information

**S1 Table. List of components with source and approximate price for constructing a single unit (as of December 2021 in Australia).**
(DOCX)

**S1 File. Instructions for building RFID readers and housing.**
(PDF)

**S2 File. Power analysis for RFID loggers.**
(PDF)

**S3 File. RFIDLOG: Dual animal tag data logger with external antenna and SD card storage.**
(PDF)

**S4 File. Northern quoll figures.**
(DOCX)

## Acknowledgments

Thanks to Northern Territory Parks and Wildlife, Kakadu National Park, Northern Land Council, the Marthakal Rangers, and Territory Wildlife Park for assistance with collection of quolls from Astell Island. Thanks to the Territory Wildlife Park for support with housing, husbandry and breeding of the quolls in captivity. Thanks to the City of Joondalup for supporting the quenda study at Craigie Bushland. Additional thanks to Robert Accardi from Priority 1 Design Pty ltd. for assistance with RFID logger set up, to Amberlee Hatcher and staff at the Territory Wildlife Park for helping to deploy the loggers, and to Ben Phillips and two anonymous reviewers for valuable comments on the manuscript.

## Author Contributions

**Conceptualization:** Natasha Dean Harrison, Ella L. Kelly.

**Data curation:** Natasha Dean Harrison, Ella L. Kelly.

**Writing – original draft:** Natasha Dean Harrison.

**Writing – review & editing:** Ella L. Kelly.

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
