## [Decision Letter · Decision Letter 0]

19 Jul 2022

PONE-D-22-12556Affordable RFID loggers for monitoring animal movement, activity and behaviour.PLOS ONE

Dear Dr. Harrison,

Thank you for submitting your manuscript to PLOS ONE. After careful consideration, we feel that it has merit but does not fully meet PLOS ONE’s publication criteria as it currently stands. Therefore, we invite you to submit a revised version of the manuscript that addresses the points raised during the review process. Please submit your revised manuscript by Sep 02 2022 11:59PM. If you will need more time than this to complete your revisions, please reply to this message or contact the journal office at plosone@plos.org. Please include the following items when submitting your revised manuscript:A rebuttal letter that responds to each point raised by the academic editor and reviewer(s). You should upload this letter as a separate file labeled 'Response to Reviewers'.A marked-up copy of your manuscript that highlights changes made to the original version. You should upload this as a separate file labeled 'Revised Manuscript with Track Changes'.An unmarked version of your revised paper without tracked changes. You should upload this as a separate file labeled 'Manuscript'.

We look forward to receiving your revised manuscript.

Kind regards,

Zhiyuan Zhu

Academic Editor

PLOS ONE

Journal Requirements:

“This work was supported by the Australian Research Council (LP150100722; FT160100198 to A/Prof Ben Phillips and A/Prof Jonathan Webb); Australian Commonwealth Government RTP Scholarship (to N.D.H); Margaret Middleton Fund Award for Endangered Australian Native Vertebrate Animals (to E.K); Hermon Slade Foundation (HSF21054 to Nicola Mitchell); and Holsworth Wildlife Research Endowment (to E.K and N.D.H).”

“This work was supported by the Australian Research Council (LP150100722; FT160100198 to A/Prof Ben Phillips and A/Prof Jonathan Webb); Australian Commonwealth Government RTP Scholarship (to N.D.H); Margaret Middleton Fund Award for Endangered Australian Native Vertebrate Animals (to E.K); Hermon Slade Foundation (HSF21054 to Nicola Mitchell); and Holsworth Wildlife Research Endowment (to E.K and N.D.H).”

“This work was supported by the Australian Research Council (LP150100722; FT160100198 to A/Prof Ben Phillips and A/Prof Jonathan Webb); Australian Commonwealth Government RTP Scholarship (to N.D.H); Margaret Middleton Fund Award for Endangered Australian Native Vertebrate Animals (to E.K); Hermon Slade Foundation (HSF21054 to Nicola Mitchell); and Holsworth Wildlife Research Endowment (to E.K and N.D.H).”

Reviewers' comments:

Reviewer's Responses to Questions

**Comments to the Author**

1. Is the manuscript technically sound, and do the data support the conclusions?

Reviewer #1: Partly

Reviewer #2: Partly

2. Has the statistical analysis been performed appropriately and rigorously? 

Reviewer #1: No

Reviewer #2: Yes

3. Have the authors made all data underlying the findings in their manuscript fully available?

Reviewer #1: No

Reviewer #2: Yes

4. Is the manuscript presented in an intelligible fashion and written in standard English?

Reviewer #1: No

Reviewer #2: Yes

5. Review Comments to the Author

Reviewer #1: Though the manuscript is interesting and the work reported is useful for researchers working in the area related to study of animal behaviour/habits. The manuscript needs improvisation on many points as explained under:

• The manuscript needs professional editing.

• Discussion part is too short, it needs to be elaborated using statistical techniques to draw detailed inferences about the species studied and performance of RFID loggers developed.

• It is not mentioned by the author that how many RFID loggers were used for Quenda study.

• Further, author should explain how and at which body part the RFID tags (PIT ) were implanted under the materials and method section.

• It is not explained, if there were any tags that didn’t work after deployment or there could be some Quenda and Quoll, which never entered the nest or antenna positioned.

• Only the data of Quenda study is correlated using the data of camera traps, the data of quoll study also needs to be explained in correlation with camera trap/other suitable techniques.

• References in the text needs to be arranged as per the format of the journal.

Reviewer #2: The work could be interesting. More technical discussion could be added as listed below.

1. More RFIDs and RFID sensors should be reviewed and discussed e.g. J Zhang, etc., A review of passive RFID tag antenna-based sensors and systems for structural health monitoring applications, Sensors, 2017.

2. The advantages of RFID monitoring systems and IOTs should be discussed.

3. More quantitative data in section 3 are expected.

6. PLOS authors have the option to publish the peer review history of their article (what does this mean?). If published, this will include your full peer review and any attached files.

Reviewer #1: No

Reviewer #2: No

---

## [Author Response · Author response to Decision Letter 0]

24 Jul 2022

We thank our two anonymous reviewers for their comments. Please see responses to each comment below.

Reviewer #1. 

Reviewer comment: Though the manuscript is interesting and the work reported is useful for researchers working in the area related to study of animal behaviour/habits. The manuscript needs improvisation on many points as explained under:

Our response: Thank you for your suggestions. We have addressed them below.

Reviewer comment: The manuscript needs professional editing.

Our response: We have not sought professional editing, instead we have had a colleague proof-read the manuscript. 

Reviewer comment: Discussion part is too short, it needs to be elaborated using statistical techniques to draw detailed inferences about the species studied and performance of RFID loggers developed.

Our response: We have now included an ANOVA in emergence time between male and female quolls (lines 162-163), an ANOVA in amount of activity between male and females quolls (lines 164-165), and estimates of within individual repeatability of both latency to emerge and foraging effort in quenda (142-144; 173-176). These results have been incorporated into the discussion (lines 197-200 and lines 223-226).

Reviewer comment: It is not mentioned by the author that how many RFID loggers were used for Quenda study.

Our response: At line 139 it reads “We deployed 5 RFID loggers over 4 nights”

Reviewer comment: Further, author should explain how and at which body part the RFID tags (PIT) were implanted under the materials and method section. 

Our response: Thanks for this suggestion. At lines 94-95 we have added “For use in small mammals, these PIT tags are inserted between the shoulder blades, on the back of the animal”

Reviewer comment: It is not explained, if there were any tags that didn’t work after deployment or there could be some Quenda and Quoll, which never entered the nest or antenna positioned.

Our response: For the quoll study, we did not have camera traps paired with each RFID reader, so we do not have this information. For the quenda study, we have clarified that “individuals were correctly identified during 25 of these (80% of visits to feeding stations; not all visits resulted in an animal entering the station)” at lines 169-170.

Reviewer comment: Only the data of Quenda study is correlated using the data of camera traps, the data of quoll study also needs to be explained in correlation with camera trap/other suitable techniques. 

Our response: Unfortunately, we did not have camera trap data for the quoll study so we are unable to correlate this information.

Reviewer comment: References in the text needs to be arranged as per the format of the journal. 

Our response: We have reformatted the references to the style of the journal. 

Reviewer #2: 

Reviewer comment: The work could be interesting. More technical discussion could be added as listed below.

Our response: Thank you for your suggestions. We have addressed them below.

Reviewer comment: 1. More RFIDs and RFID sensors should be reviewed and discussed e.g. J Zhang, etc., A review of passive RFID tag antenna-based sensors and systems for structural health monitoring applications, Sensors, 2017.

Our response: Reviewing multiple RFID sensors as Zhang et al. have done is outside the scope of our study – we are intending to evaluate a cheaper RFID option, draw comparisons to commercial technology and describe potential applications. Zhang et al review a wide range of technologies, and in this study, we are mainly focussed on short-range RFID. We have specified this in the introduction at line 66 and have made reference to the Zhang et al. paper in a broader context at line 53.

Reviewer comment: 2. The advantages of RFID monitoring systems and IOTs should be discussed.

Our response: In the discussion, we describe some advantages of using RFID technology, specifically in regard to experimental design and sample size (lines 226-237). We have also now added lines discussing IOT and the potential application at lines 238-246 “Our RFID loggers remove the need for a human observer in the field, however, they still require some human effort to collect data (downloaded manually using the Priority 1 Software). Given the rapid development of sensor technology and the modular nature of these RFID loggers, it may be possible to connect these sensors to a mobile network, where data can be received straight into the cloud (the concept of Internet Of Things; IOT [9]. Though this was outside the scope of this particular study, we recommend futures studies considering our RFID loggers take the time to investigate the potential of IOT to increase efficiency in data collection through automation. Such an advancement also opens up many avenues for further application, such as remote trapping of targeted individuals, or access-limited nest boxes based on individual identity.”.

Reviewer comment: 3. More quantitative data in section 3 are expected.

Our response: We have now included an ANOVA in emergence time between male and female quolls (lines 162-163), an ANOVA in amount of activity between male and females quolls (lines 164-165), and estimates of within individual repeatability of both latency to emerge and foraging effort in quenda (142-146; 173-176). These results have been incorporated into the discussion (lines 197-200 and lines 223-236).

---

## [Decision Letter · Decision Letter 1]

29 Aug 2022

PONE-D-22-12556R1Affordable RFID loggers for monitoring animal movement, activity, and behaviour.PLOS ONE

Dear Dr. Harrison,

Thank you for submitting your manuscript to PLOS ONE. After careful consideration, we feel that it has merit but does not fully meet PLOS ONE’s publication criteria as it currently stands. Therefore, we invite you to submit a revised version of the manuscript that addresses the points raised during the review process.

We look forward to receiving your revised manuscript.

Kind regards,

Zhiyuan Zhu

Academic Editor

PLOS ONE

Reviewers' comments:

Reviewer's Responses to Questions

**Comments to the Author**

1. If the authors have adequately addressed your comments raised in a previous round of review and you feel that this manuscript is now acceptable for publication, you may indicate that here to bypass the “Comments to the Author” section, enter your conflict of interest statement in the “Confidential to Editor” section, and submit your "Accept" recommendation.

Reviewer #1: (No Response)

Reviewer #2: All comments have been addressed

2. Is the manuscript technically sound, and do the data support the conclusions?

Reviewer #1: No

Reviewer #2: No

3. Has the statistical analysis been performed appropriately and rigorously? 

Reviewer #1: No

Reviewer #2: I Don't Know

4. Have the authors made all data underlying the findings in their manuscript fully available?

Reviewer #1: Yes

Reviewer #2: No

5. Is the manuscript presented in an intelligible fashion and written in standard English?

Reviewer #1: No

Reviewer #2: Yes

6. Review Comments to the Author

Reviewer #1: The author has tried to incorporate suggestions but changes made are still not satisfactory. The manuscripts still require professional English editing. Some of the points raised earlier are still unanswered, like data for quoll study has not been correlated with camera traps or other means. Also, author is silent about failure/damage of tags ( if any ), as asked earlier. Further, in response of application of statistical technique for detailed discussion the author has applied the ANNOVA test only, no graphical presentation/ analysis is there for data collected. While the studied animals were tagged using RFID, other data about the animals (common traits and age etc.) could have been collected for detailed study. Further foraging habits/animal activity may be linked with the vegetation or any other important variable for the habitat studied.

Manuscripts still lacks the in-depth discussion, therefore, in my view the manuscripts need to be rejected.

Reviewer #2: The addressing comments are reasonably good. More technical discussion e.g. how to use the RFID data should be provided.

7. PLOS authors have the option to publish the peer review history of their article (what does this mean?). If published, this will include your full peer review and any attached files.

Reviewer #1: No

Reviewer #2: No

---

## [Author Response · Author response to Decision Letter 1]

7 Sep 2022

We thank our two anonymous reviewers for their comments. Our overarching aim for this publication is to describe the affordable RFID readers, provide information for their construction and implementation, and to assess their performance in the field. The quoll and quenda studies are meant to act as examples of practical applications of the technology and are not the focus of the paper. For that reason, we have refrained from adding in further discussion of the specific studies and have instead focussed our discussion on the RFID technology itself, covering successes, limitations, and potential future advancements of the RFID loggers. Please see responses to each comment below.

Reviewer #1:

1. The manuscripts still require professional English editing. 

Both authors are native English speakers. We have further had two colleagues proof-read the manuscript.

2. Some of the points raised earlier are still unanswered, like data for quoll study has not been correlated with camera traps or other means. 

We did not correlate the quoll data without camera traps or another means. This point was answered in the previous review. We responded that “For the quoll study, we did not have camera traps paired with each RFID reader, so we do not have this information”. This was also added to the manuscript in the previous revision at line 211: “Although camera trap data was not able to be collected for the quoll study to ground-truth the results, camera trap footage from the quenda study showed an 80% accuracy rate.”

3. Also, author is silent about failure/damage of tags ( if any ), as asked earlier. 

None of the tags failed. This is detailed in the text at lines 165 and 176-177 that read “All RFID loggers functioned for 24 hours as expected” and “All RFID loggers functioned for 24 hours as expected” respectively.

4. Further, in response of application of statistical technique for detailed discussion the author has applied the ANNOVA test only, no graphical presentation/ analysis is there for data collected. 

Thank you for the suggestion, we have added graphical representations of these results to the Supplementary Material – S5.

5. While the studied animals were tagged using RFID, other data about the animals (common traits and age etc.) could have been collected for detailed study. Further foraging habits/animal activity may be linked with the vegetation or any other important variable for the habitat studied. 

For both studies, all animals were adults, and as they are wild, we were unable to age them in more detail than this. We were able to sex the animals and have incorporated this information into our analyses. As we were focused on trialling and validating the RFID technology, we did not collect additional data from the animals or the study site that could be incorporated into the analysis.

6. Manuscripts still lacks the in-depth discussion, therefore, in my view the manuscripts need to be rejected. 

Our aim for this paper, is to present this economical technology, provide a guide to build it and, validate its success in the field. We feel that this discussion covers the important and interesting aspects of the technology, and that we have adequately discussed the successes, limitations, and potential future uses and advancements of the RFID loggers. 

In the discussion, we have covered:

- the economic benefit of the cheaper units (including a price comparison and power analysis) at lines 187-200

- Limitations of the setup we employed, with suggestions on how to overcome them at lines 201-213

- Potential uses for this technology in ecology, including their substitution for camera traps, their use in studies of survival and dispersal, and their application in determining animal personalities (lines 214-238)

- Future technological advancements of such a setup, at lines 238-246.

Reviewer #2: 

1. The addressing comments are reasonably good. More technical discussion e.g. how to use the RFID data should be provided.

Paragraphs 4-5 on the discussion (lines 214-238) are dedicated to describing uses for the data from such studies, describing their potential use in substitution for camera traps, their use in studies of survival and dispersal, and their application in determining animal personalities.

---

## [Decision Letter · Decision Letter 2]

6 Oct 2022

Affordable RFID loggers for monitoring animal movement, activity, and behaviour.

PONE-D-22-12556R2

Dear Dr. Harrison,

We’re pleased to inform you that your manuscript has been judged scientifically suitable for publication and will be formally accepted for publication once it meets all outstanding technical requirements.

Kind regards,

Zhiyuan Zhu

Academic Editor

PLOS ONE

Additional Editor Comments (optional):

Reviewers' comments:

Reviewer's Responses to Questions

**Comments to the Author**

1. If the authors have adequately addressed your comments raised in a previous round of review and you feel that this manuscript is now acceptable for publication, you may indicate that here to bypass the “Comments to the Author” section, enter your conflict of interest statement in the “Confidential to Editor” section, and submit your "Accept" recommendation.

Reviewer #1: All comments have been addressed

Reviewer #2: All comments have been addressed

2. Is the manuscript technically sound, and do the data support the conclusions?

Reviewer #1: Partly

Reviewer #2: Yes

3. Has the statistical analysis been performed appropriately and rigorously? 

Reviewer #1: No

Reviewer #2: Yes

4. Have the authors made all data underlying the findings in their manuscript fully available?

Reviewer #1: Yes

Reviewer #2: Yes

5. Is the manuscript presented in an intelligible fashion and written in standard English?

Reviewer #1: Yes

Reviewer #2: Yes

6. Review Comments to the Author

Reviewer #1: Points raised has been addressed in the revised manuscript, so the manuscript may be considered for publication.

Reviewer #2: It is acceptable.

It would be better to discuss the disadvantages of RFID loggers e.g. Mugahid Omer, etc., Indoor distance estimation for passive UHF RFID tag based on RSSI and RCS, Measurement, Volume 127, 2018, Pages 425-430, ISSN 0263-2241, https://doi.org/10.1016/j.measurement.2018.05.116.

7. PLOS authors have the option to publish the peer review history of their article (what does this mean?). If published, this will include your full peer review and any attached files.

Reviewer #1: No

Reviewer #2: No

---

## [Editor Report · Acceptance letter]

21 Oct 2022

PONE-D-22-12556R2 

Affordable RFID loggers for monitoring animal movement, activity, and behaviour. 

Dear Dr. Harrison:

I'm pleased to inform you that your manuscript has been deemed suitable for publication in PLOS ONE. Congratulations! Your manuscript is now with our production department. 

Kind regards, 

on behalf of

Prof. Zhiyuan Zhu 

Academic Editor

PLOS ONE